# Principles and Clinical Application of Free-Style Capillary Perforator-Based Flap for Coverage of Facial Skin Cancer Defects

**DOI:** 10.3390/cancers16122206

**Published:** 2024-06-12

**Authors:** Hyung-Sup Shim, Hyun-Jung Ryoo, Jae-Seon Choi, Ji-Ah Park, Youn-Hwan Kim

**Affiliations:** 1Department of Plastic and Reconstructive Surgery, St. Vincent’s Hospital, College of Medicine, The Catholic University of Korea, Seoul 16247, Republic of Korea; hjryoo92@gmail.com (H.-J.R.); 10cjs13@naver.com (J.-S.C.); 2Design Lab of Technology Commercialization Center, Industry-University Cooperation Foundation of Hanyang University, Seoul 04763, Republic of Korea; jia11@hanyang.ac.kr; 3Department of Plastic and Reconstructive Surgery, College of Medicine, Hanyang University, Seoul 04763, Republic of Korea

**Keywords:** skin cancer, squamous cell carcinoma, basal cell carcinoma, local flap, surgery

## Abstract

**Simple Summary:**

This study introduces a groundbreaking method known as the true free-style capillary perforator flap, revolutionizing the approach to reconstructing facial skin defects, such as skin cancer. By meticulously employing a flap design technique that integrates principles such as finger-pinching and alignment with wrinkle directions, the study demonstrates successful reconstructions across various facial regions, including the infraorbital area, nose, and cheek, with a remarkable reduction in postoperative complications. Its contribution lies not only in its clinical success but also in its potential to improve patients’ quality of life by offering a reliable and aesthetically pleasing solution to facial skin defects. The true free-style capillary perforator flap stands poised to become a cornerstone in the management of facial skin defects, paving the way for enhanced patient care and outcomes in the realm of reconstructive surgery.

**Abstract:**

This study introduces a free-style perforator based island flap (PBIF) for the reconstruction of skin defects. From March 2012 to December 2022, a retrospective investigation was conducted on patients who underwent reconstruction for facial defects due to skin cancer. Data on the patients’ gender, age, anesthesia method, diagnosis, defect location, flap size, complications, and follow-up periods were collected. There are several principles for designing the PBIF: finger-pinching method, alignment with the direction of wrinkles, the smaller width and longer length of the flap, and proximal attachment to the muscle. A total of 32 patients were included, with an average age of 63.6 years. Surgeries were performed in various regions, such as the infraorbital area, nose, cheek, philtrum, and the anterior/posterior/inferior auricular regions, with an average flap size of 7.63 cm^2^. There were no complications, such as venous congestion or vascular insufficiency in the skin flaps, although one case required revisional closure due to flap disruption. The PBIF is a useful and effective method for the restoration of facial defects. This method can provide simple yet aesthetically satisfying results, showing stable outcomes without complex surgeries or complications. This study indicates the potential for this method to be more widely employed in reconstructive surgeries in the future.

## 1. Introduction

Facial defects can arise from trauma, skin cancers, and chronic illnesses. When repairing facial defects, it is essential to prioritize both function and aesthetics. Ideally, facial reconstruction should use skin and soft tissue from the facial area for optimal aesthetic and functional results. The rise in perforator-based surgeries has made flaps using perforators common in facial reconstruction. Although various flap-raising techniques using the free-style approach exist, they often do not represent a true free-style method. This approach can extend incision lines based on perforator locations, potentially using distant tissues and compromising aesthetics. Thus, we propose a genuine free-style capillary perforator flap using minimal incisions from surrounding tissue, enhancing both function and appearance. This research has not been previously published in any other journal, maintaining its originality and exclusivity in relation to prior scholarly dissemination. This work was supported by the research fund of Hanyang university.

## 2. Materials and Methods

From March 2012 to December 2022, we retrospectively investigated patients who underwent reconstruction for facial or neck defects due to skin cancer at the Department of Plastic and Reconstructive Surgery, Hanyang University Hospital, using a free-style perforator-based island flap (PBIF). We collected data on the patients’ gender, age, anesthesia method, diagnosis, defect location, flap size, complications, and follow-up periods through retrospective chart reviews. The author followed the principles of the Declaration of Helsinki, and the study was approved by the institutional review board of Hanyang University Medical Center (HY202405023). The authors obtained explicit consent from the patients for the use of data, including clinical photographs. The data were processed while preserving the confidentiality of individual identities.

### Operative Technique

Under general or local anesthesia, the skin cancer area was removed with a wide margin. Once it was confirmed by frozen biopsy that the margin was free from cancer, the flap design was initiated. After extensive removal, most defects had a circular or oval shape. The principles for designing the PBIF are as follows:The presence of sufficient tissue around the defect can be determined with a finger-pinching method.For perforator selection, it is not necessary to choose one detectable with a hand-held Doppler.The direction of the patient’s wrinkles and areas that can be hidden in the relaxed tissue around the defect should be considered.The width of the skin flap should account for the defect shape, changing from round to oval when the PBIF donor site is lifted and set around the defect.A slightly smaller width of the skin flap, about 5–10 mm, is enough to fit the defect.The length of the skin flap should be longer than the defect by about 1.0–2.0 cm because the defect changes to an oval shape.When elevating the skin flap, the proximal part of the flap should remain attached to the muscle about 1 cm in diameter without complete isolation.In the proximal part of the elevated flap, the subcutaneous layer can be sufficiently dissected to allow easy rotation of the flap.It is advantageous to design the axis of the skin flap to rotate within 90 degrees. However, considering scars or wrinkle lines, the axis of rotation can be increased up to 180 degrees.

Following primary donor site closure, the flap was sutured as two layers. A Penrose drain was inserted and patients were discharged within 4–5 days. Sutures should be removed within 7 days for optimal aesthetic outcome.

## 3. Results

A total of 32 patients were included in this study, with 19 female patients and 13 male patients. The ages of the patients ranged from 29 to 94 years, with an average age of 63.6 years. The surgery time ranged from 40 to 110 min, with an average time of about one hour (63.1 min). Initially, surgeries were performed under general anesthesia. However, as experience accumulated, eight later cases were performed under local anesthesia. Only cases of skin cancer were included in the study, with twenty cases of basal cell carcinoma, ten cases of squamous cell carcinoma, one case of sebaceous carcinoma, and one case due to follicular lymphoma of the skin. The reconstruction of the skin flap was performed in various regions, including the medial canthal region, cheek region, infraorbital region, nasolabial fold region, nose region, and behind and below the ears. The size of the skin flaps ranged from 3 × 1.5 cm to 7 × 3 cm, with an average flap size of 7.63 cm^2^. There were no complications, such as venous congestion or vascular insufficiency in the skin flaps, except for one case where the flap was disrupted and required revisional closure. In one patient who underwent reconstruction behind the ear, hypertrophic scarring occurred 2 months postoperatively, which was resolved with serial steroid injections. The average follow-up period was 31 months, and there were no cases of recurrence during this period (Table 1).

The cases presented in this study were carefully selected to effectively demonstrate the versatility and effectiveness of the free-style perforator-based island flap (PBIF) technique. The selection process focused on patients diagnosed with skin cancer. Cases were chosen to represent a variety of facial regions, highlighting the adaptability of the PBIF technique in different anatomical areas. Additionally, cases were included to showcase the PBIF’s capability in both simple and complex reconstructions. Finally, all patients included in the study provided explicit consent for the use of their clinical data and photographs.

### 3.1. Case 1

A 76-year-old male patient with a growing mass in the periorbital area on the medial canthal region. Squamous cell carcinoma was diagnosed with no metastasis found. Thus, excision was performed with sufficient margin. The frozen biopsy results confirmed the absence of cancer cells in all margins. A flap was designed along the angular artery in the supra-labial fold area, which had relatively more flexibility and could conceal the scar. To fit the round shape of the defect, the width of the flap was designed to be 1.5 cm, and the length was designed to be approximately 3.5 cm to cover the defect. To avoid significant scarring, the flap’s rotational angle was set to 160 degrees, as a rotation of 90 degrees or less was not possible. The flap was raised from the distal area at the supra-fascial level, and the deep portion of the proximal flap was preserved with a diameter of 1 mm to avoid detachment from the muscle, containing capillary perforators. Primary closure was performed. The entire operation time was 60 min, and the patient healed without complications. After six months, the scar was almost invisible (Figure 1).

### 3.2. Case 2

A 67-year-old female patient with a long-lasting mass on her nose, which was diagnosed as basal cell carcinoma on biopsy results. She underwent surgery to remove the cancerous area, which resulted in partial exposure of the nasal cartilage. Due to the complexity and size of the defect, a free-style perforator island flap was planned. To minimize scarring, a 3 × 2 cm skin flap was elevated along the nasolabial fold. Various sizes of perforators, which penetrate the muscle layer and ascend along the angular artery, were identified and dissected to include the connective tissue containing approximately one-third of the proximal perforators. The flap was raised with a rotation angle of approximately 90 degrees, allowing easy rotation. The total operation time was 70 min, including the frozen biopsy. The surgical outcome was satisfactory with no evidence of hypertrophic scarring (Figure 2).

### 3.3. Case 3

A 62-year-old female patient underwent laser treatment for an ulcer-like lesion that persisted on her cheek, but it continued to recur. Tissue examination revealed basal cell carcinoma, and she was referred to our department. The cancer was completely removed, and all frozen tests were negative. Given the prominent location on the cheek area near the lower eye and the tight surrounding tissue, a free-style perforator flap using a capillary perforator that penetrated the orbicularis oculi muscle below the eye was planned to avoid scarring above the eyebrow. A 3 × 1.5 cm flap was designed to rotate within 90 degrees of the defect, parallel to the axis of the eye, considering the direction of the muscle below the eye. The flap was raised, and the surrounding tissue was preserved to prevent detachment from the muscle at the site where the flap and the defect met. The flap was easily transferred to the defect area without excessive tension above the eye. The surgery time was 60 min, and the patient showed excellent cosmetic results without complications (Figure 3).

## 4. Discussion

Various methods have been proposed for the reconstruction of skin cancer occurring in the facial area [1,2,3]. As the face is the most directly exposed part of the body and plays the most important role in social activities, reconstruction after cancer surgery is essential to minimize scarring and provide aesthetic reconstruction [4,5]. Therefore, the reconstruction methods should prioritize considering both function and aesthetics. Ongoing discussions debate the necessity of excision beyond a certain number of centimeters, and studies are being conducted to minimize the range of excision for basal cell carcinoma and squamous cell carcinoma to prevent recurrence [6,7]. Therefore, the range of excision is gradually changing in the direction of minimization and aesthetics.

When considering aesthetic reconstruction, it is important to minimize the length of scars and to use the same tissue to rebuild, based on the “like with like” concept [8]. Recently, there has been a trend toward aesthetic reconstruction, considering the concept of the facial subunit [9,10]. When considering these situations, skin grafting can be a relatively simple method for reconstruction, but patients may not be satisfied due to problems in terms of aesthetic reconstruction [11]. Therefore, various forms of local flaps that attempt to utilize surrounding tissues have been developed, including traditional bilobed flaps, Limberg flaps, and rotation flaps [9,12,13]. These flaps can produce good results with relatively simple procedures, utilizing the surrounding tissue for reconstruction. Therefore, even for reconstructive surgeons with little experience, simple designs and surgical methods that can be applied to small to moderate-size defects can easily achieve aesthetic and functional reconstruction. However, there are problems regarding the longer incision lines than the original defect area, the risk of more visible scarring if suturing does not align with the direction of the resting skin tension line (RSTL), and the lack of applicability to relatively small and simple defects in all areas of the face. 

In recent years, as interest in perforator anatomy has continued to increase, perforator flaps have become the mainstay of reconstruction in most fields [14,15,16]. By omitting the muscle layer, there are fewer complications after surgery and lower donor site morbidity. Additionally, the characteristics of a thin skin allow for the accurately tailored resurfacing of the desired defect site, resulting in advantages in various reconstruction situations.

Research on perforators in the facial region is actively ongoing, with promising results being reported. According to several studies, relatively favorably sized perforators that can be traced by a hand-held Doppler are being investigated as they pass through the facial muscles, particularly along the angular artery, where they are distributed to the nasolabial fold region [17,18].

In addition, very small perforator maps passing through the orbicularis oculi muscle have been confirmed, which are useful for the reconstruction of periorbital defects [19]. While studying anatomical vessels distributed in the facial region, numerous networks and capillary perforators have been found to be linked and connected to each other. The concept of the plexus, initially referred to as the subdermal plexus, has evolved into the concept of capillary perforators [20,21]. These capillary perforators can be found anywhere in the facial region and can be utilized as a source vessel for true island flap elevation. In other words, any flap can be mapped using very small capillary perforators that cannot be detected by a hand-held Doppler, and can be used for defect reconstruction. In short, the use of capillary perforators in free-style perforator flaps means that the surgeon can focus on designing the flap to fit the defect area in a direction that minimizes scarring and rotation angles without excess concentration on the source perforator, decreasing the surgical time.

The propeller flap based on the perforator proposed by Hyakusoku H. often resulted in congestion at the end of the flap due to a rotation greater than 180 degrees, and the incision line was notably longer than the defect area due to the large rotation angle, resulting in a larger scar [22]. Teo TC. also proposed an evolved propeller flap concept, but it has the same problems [23], whereas the perforator-based island flap, which is a modification of this propeller flap, has a rotation angle of less than 180 degrees [24,25]. In addition, designing the flap to be attached directly to the defect area as much as possible, resulting in a rotation angle within 90 degrees, makes surgery easier and faster and reduces the incision line for less final scarring. This concept has been widely reported in the reconstruction of pressure sores on the buttocks and of female cancer surgery in the perineum [26,27].

Based on the angular artery, the authors reported a perforator-based island flap [15,28,29]. They used a Doppler to confirm the perforator around the angular artery, designed the flap based on it, then rotated the flap. However, this was limited to a modified form of the existing nasolabial fold flap, and it was difficult to apply this concept to other defect areas. However, with the development of the concept of capillary perforators, the flap can now be easily raised by free design. If the surrounding area of the flap is peeled off without separating it from the underlying muscle layer within a radius of about 10 mm, located in the proximal part of the flap, the flap can rotate easily. Since the size of the flap is not large, the primary closure of the donor site can be easily achieved by undermining the surrounding subcutaneous layer. Another advantage of capillary PBIF is a smaller incision size with less scarring and less limitation of facial expression resulting from the minimal undermining of adjacent muscle and soft tissue, providing better outcomes in social functioning aspects.

In terms of surgical time, this study showed an average of about one hour, highlighting its simplicity and ease compared to previous methods [30]. The surgical time includes the frozen biopsy time, typically lasting about 20–30 min, so the actual surgical time related to flap harvest is less than 30 min. Therefore, if the surgery is performed properly, the surgeon can easily harvest the flap with little to no complications. The proposed method can be considered a safe and easy flap procedure with minimal complications, and that is in line with the true meaning of the free-style perforator flap.

The design is the key and most important factor of the free-style perforator flap. Among the principles mentioned, when filling a defect, consideration of the ultimate oval rather than round shape, due to elevation of the flap and the area of suture, is needed. When designing the flap, the surgeon should take into account the decreased width and increased length of the defect after filling. Also, it is preferable to design the flap elevation portion so that it can be hidden within wrinkles, and it is advantageous to maintain the rotation angle within 90 degrees. As the facial defect is not large, even if there is not much surrounding tissue dissection, the flap rotates well.

This study has limitations as it is a retrospective study. Additionally, we were unable to compare it with existing surgical methods, such as skin grafts or local flaps, and the patient group is relatively small. However, the surgical method is based on a design that is not highly complex, making it easy for beginners to follow, which adds value as an educational model. Moreover, it can reduce surgery time, allows for faster patient recovery, and has shown significantly visible results from an aesthetic perspective.

## 5. Conclusions

The facial area contains numerous perforators. This allows for the easy and quick performance of the free-style capillary perforator-based island flap under local anesthesia. Free-style perforator-based island flaps are the most evolved form of propeller flap based on facial capillary perforators. The primary constraint of this capillary PBIF lies in its limited size, which led us to exclude cases of malignant melanoma and other extensive cancer defect reconstructions from the study.

## Figures and Tables

**Figure 1 cancers-16-02206-f001:**
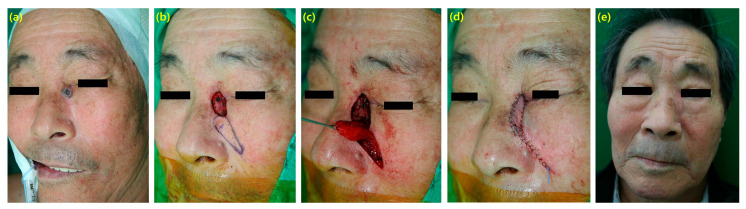
(**a**) A 76-year-old male with squamous cell carcinoma on left medial canthal area. (**b**) A 1 × 1.5 cm sized defect was shown after excision and a 1.5 × 3.5 cm sized flap was designed along the angular artery in the supra-labial fold. (**c**) The flap was raised with 160-degrees rotation for avoiding significant scarring. (**d**) Immediate postoperative view. (**e**) The outcome showed no complication.

**Figure 2 cancers-16-02206-f002:**
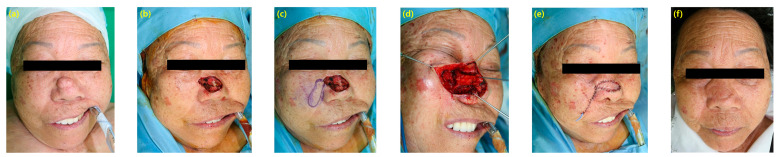
(**a**) A 67-year-old female presented with basal cell carcinoma on nasal area. (**b**) A 3 × 2 cm sized defect was shown after basal cell carcinoma excision. (**c**) A 3 × 1 cm sized free style perforator island flap was designed. (**d**) The flap was raised with a rotation angle of approximately 90 degrees, and mobilized to the defect area. (**e**) Immediate postoperative view. (**f**)The outcome showed no evidence of reconstruction or hypertrophic scarring.

**Figure 3 cancers-16-02206-f003:**
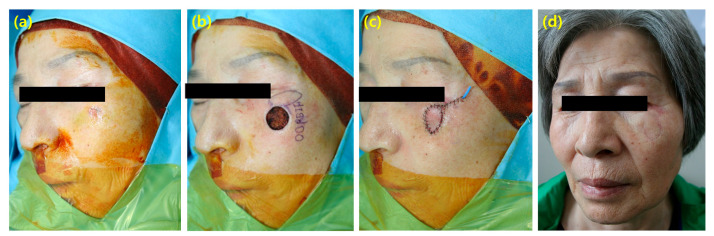
(**a**) A 62-year old female with basal cell carcinoma on left cheek. (**b**) A 1.5 × 1.5 cm sized defect was shown after basal cell carcinoma excision and a 3 × 1.5 cm sized free style perforator flap was designed. (**c**) The flap was raised with rotation within 90 degrees of the defect, parallel to the axis of the eye. (**d**) The outcome showed no complications including ectropion.

**Table 1 cancers-16-02206-t001:** Patient demographics.

	N = 32	(%)
Gender		
Male	12	37.5
Female	20	62.5
Age (years)		
Median	63.6	
Range	29–94	
Anesthesia		
General	20	62.5
Local	12	37.5
Diagnosis		
Basal cell carcinoma	20	62.5
Squamous cell carcinoma	10	31.25
Follicular lymphoma	1	3.125
Sebaceous carcinoma	1	3.125
Flap size (cm^2^)		
Median	7.63	
Range	3 × 1.5–7 × 3	
Complication		
Wound disruption	1	
Hypertrophic scar	1	
Follow-up months (median)	31	

## Data Availability

The raw data supporting the conclusions of this article will be made available by the authors on request.

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
