# Peer review of "Principles and Clinical Application of Free-Style Capillary Perforator-Based Flap for Coverage of Facial Skin Cancer Defects"

_cancers, 2024, doi:10.3390/cancers16122206_

Round 1

Reviewer 1 Report

Comments and Suggestions for Authors

Background info and clinical importance of free-style capillary perforator flap should be given. Also, in the introduction, the clinical utilization and any existing study with free-style capillary perforator flaps (marketed products or under investigation) must be given. In line number 70 it was stated that 'There are several principles for designing the PBIF' and the principles should be briefed in detail since the concept is new for many of the readers.

Is the study registered with any clinical trial database, if so the details must be given. The method can be presented as per the CONSORT 2010 Statement. Also, please check the copyright details of clinical images which utilised in the manuscript.

Overall, this manuscript is well presented and can be conder for publication after revision.

Comments on the Quality of English Language

Overall, this manuscript is well presented and can be conder for publication after revision.

Reviewer 2 Report

Comments and Suggestions for Authors

The concept of the work is interesting, but the authors should improve the flow of the manuscript with more clarity on the methods, the aim of study and the results.

1. In the Methods, please explain how you collected and analyzed your data and why you chose these particular cases to demonstrate the technique

2. In the Discussion, it is necessary to compare your experience with other similar works in literature and similar cases series

3. Limitations of the study should be acknowledged 

4. The paper is full of typos, starting from tile page with the authors' names in the title to blank spaces throughout the text, please check carefully

Comments on the Quality of English Language

Polishing of language by a native speaker is advised.

Reviewer 3 Report

Comments and Suggestions for Authors

The authors submitted a manuscript investigating the clinical application of a free-style perforator skin flap for reconstruction of facial skin cancer defects including its operational technique. Since there is need for new surgical treatment options in skin cancer in the head&neck area the authors report on a relevant topic.

The introduction provides sufficient background information. The chosen experimental techniques are appropriate. The presented results are reported in a clear and interpretable manner and align with the cited references. The discussion part and conclusion are conclusive. However there a some missing points and typing errors:

1)     Line 1-5 Authors should be separated from title

2)     Line 24: This study introduces

3)     Line 51-54 This information should be in the cover letter or in the appendix not in the introductory part

4)     Line 56: we retrospectively investigated patients, who

5)     Line 63: Delete the word “And”

6)     Line 69: What was the average size of the defects?

7)     Line 93: Patients were included

8)     Regarding complications: Were there any cases of wound infections? Was there need fpr post-operative antibiotic therapy?

Comments on the Quality of English Language

Since there are many typing errors, english editing should be performed by a mother-tongue medical writer.

Round 2

Reviewer 2 Report

Comments and Suggestions for Authors

The authors have done work to improve the manuscript, but I still feel that the something is lacking:

- how did you choose the cases to demonstrate the technique

- more confrontation with other similar studies

- the general take home message of your work and what you wanted to demonstrate

Author Response

Comments 1: how did you choose the cases to demonstrate the technique

Response 1: Thank you for pointing this out. We have added and presented the reasons for case selection in the manuscript.

Comments 2: more confrontation with other similar studies

Response 2: A comparison with other studies is provided from line 233 to line 255

Comments 3: the general take home message of your work and what you wanted to demonstrate

Response 3: the general take-home message is presented from line 273 to line 276.
